# Design and Evaluation of a Robust Observer Using Dead-Zone Lyapunov Functions—Application to Reaction Rate Estimation in Bioprocesses

**Alejandro Rincón** [1,2,*], **Fredy E. Hoyos** [3] **and Gloria M. Restrepo** [2]

1   Grupo de Investigación en Desarrollos Tecnológicos y Ambientales–GIDTA, Facultad de Ingeniería y Arquitectura, Universidad Católica de Manizales, Carrera 23 N. 60-63, Manizales 170002, Colombia

2   Grupo de Investigación en Microbiología y Biotecnología Agroindustrial—GIMIBAG, Instituto de Investigación en Microbiología y Biotecnología Agroindustrial, Facultad de Ciencias de la Salud, Universidad Católica de Manizales, Carrera 23 N. 60-63, Manizales 170002, Colombia; grestrepo@ucm.edu.co

3   Departamento de Energía Eléctrica y Automática, Facultad de Minas, Universidad Nacional de Colombia, Sede Medellín, Carrera 80 No. 65-223, Robledo, Medellín 050041, Colombia; fehoyosve@unal.edu.co

*   Correspondence: arincons@ucm.edu.co; Tel.: +57-(606)-8933050

**Abstract:** This paper addresses the design and evaluation of a robust observer for second order bioprocesses considering unknown bounded disturbance terms and uncertainty in the dynamics of the unknown and known states. The observer design and the stability analysis are based on dead-zone Lyapunov functions, and a detailed procedure is provided. The transient response bounds and the convergence region of the unknown observer error are determined in terms of the disturbance bounds, considering persistent but bounded disturbances in the dynamics of both the known and unknown observer errors. This is a significant contribution to closely related observer design studies, in which the transient response bounds are determined, but persistent and bounded disturbances are not considered in the dynamics of the known observer error. Other important contributions are: (i) the procedure for defining the observer parameters is significantly simpler than common observer designs, since a solution to the Ricatti equation, solution to LMI constraints, or the accomplishment of eigenvalue inequality conditions are not required; (ii) discontinuous signals are not used in the observer; and (iii) the effect of the gain sign associated with the unknown state in the dynamics of the known state is explicitly and clearly considered in the observer design and in the convergence study. In addition, the guidelines for selecting the observer parameters are provided. Numerical simulation confirms the stability analysis results: the observer errors converge within a short time, with a low estimation error, if observer-parameters are properly defined.

**Keywords:** state estimation; nonlinear observer design; nonlinear systems; bioprocess monitoring; software sensor

## 1. Introduction

Due to the adverse influence of anthropogenic activities on world climate stress, the scarcity of natural resources and environmental deterioration, development agendas worldwide have aligned strategies to rapidly migrate to a bioeconomy, where renewable resources can be transformed into food/feed, chemicals, materials, or energy. However, the viability of these technoeconomic bioprocesses is limited by relevant challenges such as composition variability in renewable raw materials, sensitivity to biological transformations and complex dynamics and monitoring difficulties, which lead to a lack of process understanding and high-fidelity models, thus limiting the capability of process optimization and slowing technological breakthroughs [1]. From a process and control perspective, the application of advanced processes is constrained due to the unknown time varying nature of influent

substrate characteristics and biological systems (relevant for second generation biorefineries). The reason is that uncertainty sources generate uncertainty in biomass formation rate, substrate consumption rate and product formation rate [2,3].

To overcome the monitoring limitations, state observers can be used to provide improved monitoring useful for system diagnosis, adaptive optimization, and model-based bioprocess control. For instance, nonlinear state observers can be used to estimate biochemical reaction terms [4,5]. Furthermore, sliding mode and super twisting observers (STO) are capable of guaranteeing convergence of the estimation error towards zero or a small value, while accounting for disturbance terms, which can be caused by unmodelled variations, unknown model terms or measurement noise [6–9]. In addition, STOs achieve fast and finite time convergence [2,10,11]. In [2], a design procedure for a super twisting observer is developed for second order systems, and guidelines for the selection of the observer parameters are proposed, considering assumptions related to the uncertainty boundaries. The developed approach is applied for the estimation of growth and uptake rate in a photo-bioreactor. In [12], a high gain observer is formulated for a fed-batch process of ethanol production by Saccharomyces cerevisiae, where starch is converted to glucose (by enzymatic hydrolysis) and glucose is consumed for both biomass growth and ethanol production. In that case, the concentrations of starch and glucose are known. The observer estimates the reaction rate of enzymatic hydrolysis and biomass growth rate, based on the measurements of starch and glucose concentrations. In [13], a second order sliding mode observer is proposed for a continuous process, for the estimation of microbial growth rate and biomass concentration, based on a known product concentration. The convergence region of the biomass observer error is determined in terms of the upper bound of the uncertainty as a function of either the growth rate or biomass concentration. Simulation is performed for a batch fermentation of lactic acid bioproduction. In [14], an extremum seeking scheme is proposed for optimization of the specific growth rate (SGR) in fed-batch processes, using a high order sliding mode observer for estimating the SGR, based on measurements of biomass concentration.

In [15], a hybrid observer is proposed for high-cell density culture of *S. Cerevisiae*, and it combines an asymptotic and an extended Kalman filter observer. The observer estimates the concentrations of biomass, ethanol and specific growth rate, using the measurements of dissolved oxygen, carbon dioxide and glucose. The culture experiences switch between oxidative and respiro-fermentative regimes under aerobic conditions. Simulations show that the observer is capable of estimating the specific growth rate during different metabolic regimes and during metabolic switch, and also exhibits the following features: (i) adequate stability and convergence under various measurement noises and parametric uncertainty, and (ii) higher performance compared to asymptotic observer and extended Kalman filter. In addition, the metabolic switch is computed on the basis of the growth rate estimation and the critic value. In [16], an observer is designed for a bioconversion process, consisting of conversion of methane to lactate by bacterium *Methylomicrobium buryatense 5GB1*. The model consists of six mass flow balance equations, the state variables of which are the concentrations of biomass (X), $CH_4$ in the gas phase ($S_{CH4,G}$), $CH_4$ in the liquid phase ($S_{CH4,L}$), $O_2$ in the liquid phase ($S_{O2,L}$), $O_2$ in the gas phase ($S_{O2,G}$), lactate in the liquid phase ($P_{lact,L}$). Eight different configurations of measured states are considered, and the observability analysis indicates that all of these configurations are observable. Extended and unscented Kalman filters are designed to account for the nonlinearity of the system. Additionally, practical observability was assessed, using the empirical observability Graminan, and it was concluded that using biomass measurement decreases practical observability due to its measurement noise. Simulations show the convergence of the observer errors in the presence of measurement noise. In [17], a robust fuzzy state observer is designed for a nonlinear system with input quantization, unknown control directions and unknown external disturbances. The unknown nonlinear terms are approximated by fuzzy logic systems. A coordinate transformation is applied so that the control gains of the transformed system are known. The unknown system states are estimated by the fuzzy

observer, and the external disturbances are estimated by the disturbance observer. The boundedness of all signals and the convergence of the observer errors are ensured. Finally, the observer is applied to an isothermal continuous stirred tank reactor by simulation.

In the observer applications, the capability of the observers to overcome the effect of disturbances depends on the disturbance features considered in the observer design, which is based on the dynamics of the observer error between the states and their estimates. Several nonlinear biosystems are represented by second order models or can be recasted to this form with a known and an unknown state (designed by $x_1$ and $x_2$). An observer for this system involves states $\hat{x}_1$, $\hat{x}_2$ which are the estimates of $x_1$, $x_2$; and the observation errors for the known and unknown states are $\overline{x}_1 = \hat{x}_1 - x_1$, $\overline{x}_2 = \hat{x}_2 - x_2$, respectively. The resulting dynamic model of the observation errors involves disturbance terms $\delta_1$ and $\delta_2$, respectively. The convergence of the observation errors depends on the considered properties of the disturbance terms:

- When the additive disturbance term $\delta_1$ is zero, the system is observable and the observer errors converge to zero, even if the disturbance term $\delta_2$ is different from zero. The convergence of the observer states for this case is determined in [10,11].
- In other cases, it is assumed that the disturbance terms $\delta_1$, $\delta_2$ are upper bounded by injection nonlinearities, which are a function of the observer errors and are later used as stabilizing functions in the observer model. This assumption implies that $\delta_1$ vanishes when the observer error $\overline{x}_1$ vanishes, but it is nonzero otherwise. This case is studied in [2,10].
- When the disturbance term $\delta_1$ is persistent but bounded, the observer errors converge to the origin neighborhood, but not to zero [2,10]. The stability for this case is known as "practical stability" and is studied in [18]. Therein, it was established that the observer errors converge to some compact set if the observer parameters are properly defined, but the size of the convergence region is not determined.

For larger systems, an observer study for a third order model is addressed in [13], where the disturbance $\delta_1$ is persistent, and the convergence of the biomass observer error is analyzed, as the biomass observer error is a linear filter of the observer error $\overline{x}_2$. However, the stability analysis assumes that the observer error $\overline{x}_1$ and its time derivative vanish, and the disturbance of the second observer error dynamics is required to be lower than a constant value that is associated with eigenvalue conditions.

In addition to the determination of the convergence region of the observer errors, it is also convenient to avoid discontinuous signals and to determine the bound of the transient response of the observer errors since:

- Discontinuous signals are commonly used in observers, for instance in [2,13]. Those signals have the following drawbacks: (i) it may lead to the possible failure of the trajectory's unicity and introduce the need to use Filippov's construction in case of sliding motion [19]; and (ii) a numerical solution to the differential equations may be problematic [2]. To mitigate these problems, saturation type signals can be introduced, which are commonly used in robust control design to avoid input chattering [20].
- Determining the bound of the transient response of the observer errors allows estimating the convergence speed, but also the effect of model uncertainties, user-defined parameters and initial values of observer states on the convergence. In turn, it allows drawing guidelines for setting the observer parameters and the initial values of the observer states [21].

In the numerical solutions to differential equations with discontinuous right-hand side, traditional numerical methods may become inaccurate or inefficient, even if the state trajectory does not stay in the discontinuity. Indeed, the derivatives are not computed correctly if the switching time is not accurately identified. Therefore, an adequate numerical approach intended for discontinuous differential equations must be used [22,23].

In this paper, the design of a robust observer for second order biosystems is proposed and evaluated and its convergence is investigated considering unknown bounded distur-

bance terms in the dynamics of the observer errors. In particular, no model of the unknown state is required. The observer design and the stability analysis are based on the theory of dead-zone Lyapunov functions. Using this approach, it is guaranteed that the estimation error of the known state converges to a compact set where its width is user-defined. In addition, guidelines for selection of the user-defined observer parameters are provided. At last, the observer is applied to estimate the reaction rate terms of the bioreactor.

The main contribution of the developed observer study is that the transient response boundary and the convergence region of the unknown observer error are determined in terms of the bounds of the disturbance terms, considering persistent but bounded disturbances in the dynamics of both the known and unknown observer errors. In contrast: (i) the disturbance of the dynamics of the known observer error is assumed to be zero in [24] and not persistent in [10,11], and in the non-practical stability case in [2]; and (ii) only the convergence region of the known observer error is determined in the practical stability case in [2]. Furthermore, other contributions compared with observer designs in the literature for systems with bounded disturbance terms are:

- The procedure to define the observer parameters is simpler. Commonly, a solution to the Ricatti equation (see [21]), solution to LMI constraints (see [2]), and accomplishment of eigenvalue inequality conditions (see [13]) are required. In contrast, those procedures are not required in the presented observer; thus, the trial-and-error effort (or sensitivity-based approaches) for defining the observer parameters is significantly reduced.
- The effect of the gain sign of the unknown state in the dynamics of the known state is explicitly and clearly considered in the stability analysis and the observer design, whereas this is lacking in observers for general structure (for instance [2]).
- Discontinuous signals are not used, while signum-type signals are commonly used (see [2,13]).

The paper is organized as follows. The second order generic model is presented in Section 2. The observer equations and the results of the observer design and stability analysis are depicted in Section 3, whereas the details are presented in the Supplementary Material. The application of the observer to bioreactors and the numerical simulation are shown in Section 4. Finally, the conclusions are drawn in Section 5.

## 2. Dynamic Model

Consider the system

$$\frac{dx_1}{dt} = h_1 + bx_2 + \delta_1 \tag{1}$$

$$\frac{dx_2}{dt} = h_2 + \delta_2 \tag{2}$$

where $x_1$, $x_2$ are the states; $h_1$, $h_2$ are model functions; $\delta_1$, $\delta_2$ are disturbance terms; and $b$ is the $x_2$ gain in the dynamics of $x_1$. The model terms fulfill the following assumptions:

**Assumption 1.** The functions $h_1$, $h_2$ are known; the state $x_1$ is measured and the coefficient $b$ is known; the state $x_2$ and the terms $\delta_1$, $\delta_2$ are unknown.

**Assumption 2.** The coefficient $b$ is bounded away from zero:

$$|b| \geq b_{\min} > 0 \tag{3}$$

where $b_{\min}$ is an unknown positive constant.

**Assumption 3.** The disturbance term $\delta_1$ is persistent but bounded:

$$\delta_1 \neq 0, \ |\delta_1| \leq d_1 \tag{4}$$

where $d_1$ is an unknown positive constant and the disturbance term $\delta_2$ is bounded.

**Remark 1.** *The state $x_2$ is the state to be estimated. The case of unknown dynamics of the unknown state corresponds to $h_2 = 0$.*

**Remark 2.** *Notice that the gain b is not required to be constant, whereas the disturbance terms $\delta_1, \delta_2$ are not constrained to be functions or noise.*

**Remark 3.** *When $\delta_1 = 0$, it is posible to guarantee convergence of the estimation error $\overline{x}_2 = \hat{x}_2 - x_2$ to zero. However, when $\delta_1$ is persistent but bounded according to (4), only "practical stability" is achieved. That is, it is guaranteed that [2,10]:*

(i)    *The error $\overline{x}_2 = \hat{x}_2 - x_2$, the estimation error of the unknown state, converges to the origin neighborhood;*
(ii)   *The width of the convergence region of $\overline{x}_2$ depends on $d_1$, the bound of the disturbance $\delta_1$ ;*
(iii)  *The width of the convergence region of $\overline{x}_2$ can be reduced to some extent, but it cannot be made arbitrarily small; this is because of $\delta_1$.*

### 3. Observer Algorithm, Observer Design and Stability Analysis

This section includes: (i) the observer equations in Sub Section 3.2; (ii) the results of the observer design and the stability analysis in Sub Section 3.2, including the transient response bounds and the convergence region of the unknown observer error. To this end, the generic second order model (Equations (1) and (2)), subject to assumptions 1 to 3, is considered. The details of the stability analysis are presented in the Supplementary Material.

*3.1. Observer Equations*

The observer equations are:

$$\frac{d\hat{x}_1}{dt} = b\hat{x}_2 - |b|\left(\omega\overline{x}_1 + \left(k + \frac{1}{4\omega}\right)\psi_{x_1} + sat_{x_1}\hat{\theta}_\delta\right) + h_1 \tag{5}$$

$$\frac{d\hat{x}_2}{dt} = -b\omega\left(\left(k + \frac{1}{4\omega}\right)\psi_{x_1} + sat_{x_1}\hat{\theta}_\delta\right) + h_2 \tag{6}$$

$$\frac{d\hat{\theta}_\delta}{dt} = \gamma|b||\psi_{x_1}| \tag{7}$$

where

$$\overline{x}_1 = \hat{x}_1 - x_1 \tag{8}$$

$$\psi_{x_1} = \begin{cases} \overline{x}_1 - \varepsilon & \text{for} & \overline{x}_1 \geq \varepsilon \\ 0 & \text{for} & \overline{x}_1 \in [-\varepsilon,\ \varepsilon] \\ \overline{x}_1 + \varepsilon & \text{for} & \overline{x}_1 \leq -\varepsilon \end{cases} \tag{9}$$

$$sat_{x_1} = \begin{cases} 1 & \text{for} & \overline{x}_1 \geq \varepsilon \\ \frac{1}{\varepsilon}\overline{x}_1 & \text{for} & \overline{x}_1 \in [-\varepsilon,\ \varepsilon] \\ -1 & \text{for} & \overline{x}_1 \leq -\varepsilon \end{cases} \tag{10}$$

$$\sigma = sign(b)$$

In addition, the observer model, $\hat{x}_1$ is the estimate of $x_1$, $\hat{x}_2$ is the estimate of $x_2$, $\hat{\theta}_\delta$ is the updated parameter, and: (i) $\gamma, k, \omega$ are user-defined positive constants; (ii) the width of the convergence region of $\overline{x}_1$, that is, $\varepsilon$, is user-defined, positive and constant, thus it is independent of model parameters or bounds of model terms. The observer structure is shown in Figure 1.

**Figure 1.** General structure of the observer. $x_1$ is the known state, $x_2$ is the unknown state, $b$, $h_1$, $h_2$ are known terms of plant model (1), (2), and $\hat{x}_2$ is the estimate of $x_2$.

**Remark 4.** *The formulated observer equations use saturation instead of discontinuous signals, whereas the bounded nature of the updated parameter $\hat{\theta}_\delta$, the asymptotic convergence of the observer error $\overline{x}_2 = \hat{x}_2 - x_2$ are ensured in the observer design and stated in Theorem 3, being $\Omega_{x2} = \{\overline{x}_2 : |\overline{x}_2| \leq \max\{-\delta_{min}, \delta_{max}\} + \omega\varepsilon\}$ the convergence set, where $\delta_{min}$, $\delta_{max}$ are unknown constants that satisfy $\delta \geq \delta_{min}$, $\delta_{min} \in (-\infty, 0]$, $\delta \leq \delta_{max}$, $\delta_{max} \in [0, \infty)$, $\delta = \frac{1}{b}\left(\frac{\delta_2}{\sigma\omega} - \delta_1\right)$. To this end, dead-zone Lyapunov functions are properly defined and applied.*

**Remark 5.** *(Guidelines for the choice of the observer parameters.) To achieve proper convergence speed and width of the convergence region of the unknown observer error $\overline{x}_2$, it is convenient to use the following observer parameters:*

*(i) a high positive value of $k$ that leads to proper convergence rate of $\overline{x}_1$;*

*(ii) a low positive value of $\varepsilon$, a value of $\hat{x}_{1|to}$ fulfilling $\hat{x}_{1|to} \in \left[-\varepsilon + x_{1|to}, \varepsilon + x_{1|to}\right]$, and a high positive value of $\gamma$, to reduce the bound of the transient response of $\overline{x}_2$;*

*(iii) a positive value of $\omega$ that gives a balance between convergence speed and width of the convergence region for $\overline{x}_2$.*

**Remark 6.** *In the application of the developed observer, the case of unknown dynamics of the unknown state can be addressed by using $h_2 = 0$.*

*3.2. Observer Design and Stability Analysis*

The observer design is based on dead-zone Lyapunov functions; this is an interesting approach that allows achieving convergence of the observer states to compact sets, despite unknown disturbance terms, while avoiding the use of discontinuous signals. Dead-zone Lyapunov functions have been mainly applied to control design: early global stability studies are presented in [20,25,26], and recent studies in [27–32]. Additionally, there are a few applications for stability analysis of open loop systems, for instance [33,34].

The observer design procedure includes the following tasks: definition of the general observer structure; definition of the observer errors $\overline{x}_1 = \hat{x}_1 - x_1$, $\overline{x}_2 = \hat{x}_2 - x_2$, and the weighted sum $z$; definition of the subsystem Lyapunov function $V_z$ corresponding to $z$ and determination of the dynamics of $z$ and $V_z$; definition of the subsystem Lyapunov function $V_{x1}$ corresponding to $\overline{x}_1$, and determination of the dynamics of $\overline{x}_1$ and $V_{x1}$; selection of the observer terms in accordance with the time derivatives of $V_z$ and $V_{x1}$; determination of the convergence properties of $z$, $\overline{x}_1$ and $\overline{x}_2$.

**Theorem 1.** *(Convergence of the weighted sum of the observer errors.) Consider the model (1), (2) subject to assumptions 1 to 3 and the observer (5)–(7), with definitions (8)–(10) and observer error $\overline{x}_2 = \hat{x}_2 - x_2$. As a result of this observer:*

*(Ti) the function $z = \overline{x}_2 - \sigma\omega\overline{x}_1$, $\sigma = sign(b)$, satisfies $\frac{dz}{dt} = (-1)\omega|b|(z + \delta)$, where $\delta = \frac{1}{b}\left(\frac{\delta_2}{\sigma\omega} - \delta_1\right)$;*

*(Tii) the function $z$ converges to $\Omega_z = \left[z^l, z^u\right]$, where $z^l = -\delta_{max} \leq 0$, $z^u = -\delta_{min} \geq 0$, and $\delta_{min}$, $\delta_{max}$ are unknown constants that satisfy $\delta \geq \delta_{min}$, $\delta_{min} \in (-\infty, 0]$, $\delta \leq \delta_{max}$, $\delta_{max} \in [0, \infty)$;*

*(Tiii) the upper bound of the transient response of $z$ is*

$$|z| \leq \left|\psi_{z|to}\right| e^{-\omega b_{min}(t-t_0)} + max\{-\delta_{min}, \delta_{max}\}.$$

**Proof.** Consider the Lyapunov function

$$V_z = \frac{1}{2}\psi_z^2$$

$$\psi_z = \begin{cases} z + \delta_{min} & \text{for} & z \geq -\delta_{min} \geq 0 \\ 0 & \text{for} & z \in (-\delta_{max}, -\delta_{min}) \\ z + \delta_{max} & \text{for} & z \leq -\delta_{max} \leq 0 \end{cases}$$

where $z = \bar{x}_2 - \sigma\omega\bar{x}_1$, $\sigma = sign(b)$, $\bar{x}_1 = \hat{x}_1 - x_1$, $\bar{x}_2 = \hat{x}_2 - x_2$,

$$\delta = \frac{1}{b}\left(\frac{\delta_2}{\sigma\omega} - \delta_1\right)$$

Additionally, $\delta_{min}$, $\delta_{max}$ are unknown constants that satisfy

$$\delta \geq \delta_{min}, \ \delta_{min} \in (-\infty, 0], \ \delta \leq \delta_{max}, \ \delta_{max} \in [0, \infty)$$

Additionally, $\hat{x}_1$, $\hat{x}_2$ are provided by the general observer form

$$\frac{d\hat{x}_1}{dt} = b\hat{x}_2 - bg_1 + h_1$$

$$\frac{d\hat{x}_2}{dt} = -bg_2 + h_2$$

$$g_1 = \sigma\omega\bar{x}_1 + \frac{1}{\sigma\omega}g_2$$

where $h_1$, $h_2$, $b$ are terms of model (1), (2) and $g_2$ is a function that will be defined later. Differentiating $z$ and the Lyapunov function $V_z$ with respect to time, arranging and using the definitions of $V_z$ and $\psi_z$, yields

$$\frac{dz}{dt} = (-1)\omega|b|(z + \delta)$$

$$\frac{dV_z}{dt} \leq -2\omega|b|V_z \leq -2\omega b_{min}V_z \leq 0$$

$$|\psi_z| \leq \left|\psi_{z|to}\right|e^{-\omega b_{min}(t-t_0)}$$

$$|z| \leq \left|\psi_{z|to}\right|e^{-\omega b_{min}(t-t_0)} + \max\{-\delta_{min}, \delta_{max}\}$$

In addition, $\psi_z$ converges to zero and $z$ converges to $\Omega_z = \left[z^l, z^u\right]$, $z^l = -\delta_{max} \leq 0$, $z^u = -\delta_{min} \geq 0$. Thus, statements Ti, Tii and Tiii are accomplished. □

**Remark 7.** *The convergence region of $z$, that is $\Omega_z$, depends on the bounds of $\delta$, hence on the bounds of $\delta_2/(b\omega)$ and $\delta_1/b$. Consequently, its width can be reduced to some extent by choosing a high $\omega$ value, but it cannot be made arbitrarily small.*

**Remark 8.** *The convergence rate of $z$ is given by the $dz/dt$ expression and the definition of $\delta$: (i) large values of $\omega$ increase the convergence rate of $z$; (ii) large values of $\omega$ decrease the effect of disturbance $\delta_2$, but not the effect of disturbance term $\delta_1$.*

**Theorem 2.** *(Convergence of $\bar{x}_1$ and boundedness of the updated parameter.) Consider the model (1), (2) subject to assumptions 1 to 3 and the observer (5)–(7), with definitions (8)–(10). As a result of this observer: (Ti) the updated parameter $\hat{\theta}_\delta$ remains bounded; (Tii) the observer error $\bar{x}_1 = \hat{x}_1 - x_1$ asymptotically converges to $\Omega_{x1} = [-\varepsilon, \ \varepsilon]$; (Tiii) the bound of the transient response*

*of $\bar{x}_1$ is: $|\bar{x}_1| \leq \varepsilon + \sqrt{\psi_{x1|to}^2 + \psi_{z|to}^2 + \gamma^{-1}\tilde{\theta}_{to}^2}$, where $\psi_{x1|to}$, $\psi_{z|to}$, $\tilde{\theta}_{to}$ are the initial values of $\psi_{x1}$, $\psi_z$, $\tilde{\theta}_\delta$, respectively, $\psi_{x1}$ is given by*

$$\psi_{x1} = \begin{cases} \bar{x}_1 - \varepsilon & for & \bar{x}_1 \geq \varepsilon \\ 0 & for & \bar{x}_1 \in [-\varepsilon, \varepsilon] \\ \bar{x}_1 + \varepsilon & for & \bar{x}_1 \leq -\varepsilon \end{cases},$$

*$\tilde{\theta}_\delta$ is $\tilde{\theta}_\delta = \hat{\theta}_\delta - \theta_\delta$, and $\theta_\delta$ is the upper bound of $(-\delta_{zt} - \delta_1/b)$.*

**Proof.** Consider the $g_2$ function and the subsystem Lyapunov functions:

$$g_2 = \omega\left(\left(k + \frac{1}{4\omega}\right)\psi_{x1} + sat_{x1}\hat{\theta}_\delta\right),$$

$$V_{z\theta x1} = V_{zx1} + V_\theta,$$

$$V_\theta = \frac{1}{2}\gamma^{-1}\tilde{\theta}_\delta^2$$

$$V_{zx1} = V_{x1} + V_z$$

$$V_{x1} = \frac{1}{2}\psi_{x1}^2$$

where

$$\psi_{x1} = \begin{cases} \bar{x}_1 - \varepsilon & for & \bar{x}_1 \geq \varepsilon \\ 0 & for & \bar{x}_1 \in [-\varepsilon, \varepsilon] \\ \bar{x}_1 + \varepsilon & for & \bar{x}_1 \leq -\varepsilon \end{cases}$$

$$\frac{d\hat{\theta}_\delta}{dt} = \gamma|b||\psi_{x1}|$$

$$\tilde{\theta}_\delta = \hat{\theta}_\delta - \theta_\delta$$

$$\delta_{zt} = \psi_z - z$$

$\theta_\delta$ is the upper bound of $(-\delta_{zt} - \delta_1/b)$, and it is unknown, positive, and constant: $|-\delta_{zt} - \delta_1/b| \leq \theta_\delta$. Differentiating $V_{z\theta x1}$ with respect to time and arranging, integrating, and applying the Barbalat's lemma, yields

$$\frac{dV_{z\theta x1}}{dt} \leq -kb_{min}\psi_{x1}^2 \leq 0$$

$$|\bar{x}_1| \leq \varepsilon + \sqrt{\psi_{x1|to}^2 + \psi_{z|to}^2 + \gamma^{-1}\tilde{\theta}_{to}^2}$$

$$\lim_{t \to \infty} \psi_{x1}^2 = 0$$

Therefore, $\bar{x}_1$ converges asymptotically to $\Omega_{x1} = [-\varepsilon, \varepsilon]$, and $\tilde{\theta}_\delta \in \mathcal{L}_\infty$, $\hat{\theta}_\delta \in \mathcal{L}_\infty$. Thus, statements Ti, Tii and Tiii are accomplished. □

**Theorem 3.** *(Convergence of $\bar{x}_2$: upper bound of the transient response and convergence region.) Consider the model (1), (2) subject to assumptions 1 to 3 and the observer (5)–(7), with definitions (8)–(10). As a result of this observer: (Ti) the transient response of the observer error $\bar{x}_2 = \hat{x}_2 - x_2$ satisfies:*

$$|\bar{x}_2| \leq \left|\psi_{z|to}\right|e^{-\omega b_{min}(t-t_0)} + \max\{-\delta_{min}, \delta_{max}\} + \omega\left(\varepsilon + \sqrt{\psi_{x1|to}^2 + \psi_{z|to}^2 + \gamma^{-1}\tilde{\theta}_{to}^2}\right),$$

*where* $\delta = \frac{1}{b}\left(\frac{\delta_2}{\sigma\omega} - \delta_1\right)$; *(Tii)* $|\overline{x}_2| \leq \left|\psi_{z|to}\right|e^{-\omega b_{min}(t-t_0)} + \max\{-\delta_{min}, \delta_{max}\} + \omega\left(\varepsilon + \left|\psi_{z|to}\right|\right)$

*holds true for* $\hat{x}_{1|to} \in \left[-\varepsilon + x_{1|to}, \ \varepsilon + x_{1|to}\right]$ *and a high* $\gamma$ *value leading to* $\gamma^{-1}\widetilde{\theta}_{to}^2 \approx 0$; *(Tiii) the observer error* $\overline{x}_2$ *asymptotically converges to*

$$\Omega_{x2} = \{\overline{x}_2 : \ |\overline{x}_2| \leq \max\{-\delta_{min}, \delta_{max}\} + \omega\varepsilon\}.$$

**Proof.** From the definition of $z$, and the results of convergence of $z$ and $\overline{x}_1$ obtained in Theorem 1 and Theorem 2, it follows that $\overline{x}_2$ fulfills:

$$|\overline{x}_2| \leq \left|\psi_{z|to}\right|e^{-\omega b_{min}(t-t_0)} + \max\{-\delta_{min}, \delta_{max}\} + \omega\left(\varepsilon + \sqrt{\psi_{x1|to}^2 + \psi_{z|to}^2 + \gamma^{-1}\widetilde{\theta}_{to}^2}\right)$$

$$|\overline{x}_2| \leq \left|\psi_{z|to}\right|e^{-\omega b_{min}(t-t_0)} + \max\{-\delta_{min}, \delta_{max}\} + \omega\left(\varepsilon + \left|\psi_{z|to}\right|\right)$$

for $\hat{x}_{1|to} \in \left[-\varepsilon + x_{1|to}, \ \varepsilon + x_{1|to}\right]$ and a high $\gamma$ value leading to $\gamma^{-1}\widetilde{\theta}_{to}^2 \approx 0$.

Additionally, $\overline{x}_2$ converges to

$$\Omega_{x2} = \{\overline{x}_2 : \ |\overline{x}_2| \leq \max\{-\delta_{min}, \delta_{max}\} + \omega\varepsilon\}$$

where $\psi_{x1|to}$, $\psi_{z|to}$, $\widetilde{\theta}_{to}$ are the initial values of $\psi_{x1}$, $\psi_z$, $\widetilde{\theta}_b$. Thus, statements Ti, Tii and Tiii are accomplished. $\square$

**Remark 9.** *The bound of the transient response of* $\overline{x}_2$ *depends on model terms* $b$, $\delta_2$, $\delta_1$ *and user-defined parameters* $\omega$, $\varepsilon$, $\gamma$. *Indeed, the bound of the transient response of* $\overline{x}_2$ *can be decreased by using a low value of* $\varepsilon$ *and the conditions:* $\hat{x}_{1|to} \in \left[-\varepsilon + x_{1|to}, \ \varepsilon + x_{1|to}\right]$ *and a high* $\gamma$ *value leading to* $\gamma^{-1}\widetilde{\theta}_{to}^2 \approx 0$. *In addition, the convergence speed can be increased by using a high* $\omega$ *value, but it would also increase the width of the convergence region.*

**Remark 10.** *The convergence region of* $\overline{x}_2$, *that is* $\Omega_{x2}$, *depends on model terms* $b$, $\delta_2$, $\delta_1$ *and user-defined parameters* $\omega$, $\varepsilon$. *Additionally, it can be reduced to some extent by using a low* $\varepsilon$ *value and by properly defining* $\omega$, *but it cannot be made arbitrarily small, due to the presence of* $\delta_2$, $\delta_1$, *and the effect of* $\omega$.

**Remark 11.** *The bound of the transient response and the convergence region of* $\overline{x}_2$ *are not affected by parameter* $k$. *However, the convergence of* $\overline{x}_1$ *is strongly affected by parameter* $k$, *as can be concluded from the results from Theorem 2.*

**Remark 12.** *The convergence of* $\overline{x}_1$ *is asymptotic and it depends on user-defined gains* $k$, $\gamma$. *Additionally, its convergence can be improved through a high* $k$ *value.*

**Remark 13.** *The guidelines provided in Remark 5 are derived from Remarks 9 to 12.*

*3.3. Discussion of Observer Design and Evaluation Results*

The robust observer has been designed for a second order system involving bounded persistent disturbance terms. The bound of the transient response and the convergence region of the unknown observer error have been determined in terms of the bounds of persistent disturbance terms. Additionally, the guidelines for the choice of the observer parameters have been provided in Remark 5.

Some improvements have been made in the observer design procedure in order to achieve the aforementioned contributions of the work, namely: (Ti) the use of dead-zone modifications in the definition of the subsystem Lyapunov functions $V_{x1}$ and $V_z$; (Tii) the definition of the weighted sum of the observer errors ($z$), involving the sign of $b$; (Tiii) the use of $z$ instead of $\overline{x}_2$ in the definition of the Lyapunov function; (Tiv) the consideration of the persistent disturbance term $\delta_1$ in the dynamics of $\overline{x}_1$. The basic idea of improvements Iii

and Iiii is taken from [35], but in this work this idea has been developed using dead-zone Lyapunov functions and the sign of $b$ has been incorporated in the definition of $z$, and also improvement Iiv has been made.

The improvements (Ti) and (Tiii) allow us to simplify the observer design, such that the solution to the Ricatti equation, solution to LMI constraints and the fulfillment of eigenvalue conditions are not necessary. Indeed, the use of the dead-zone Lyapunov function facilitates the examination of the convergence to compact sets, but it also allows us to avoid using discontinuous signals in the update law and in the observer equations.

The equations of the transient response and convergence region of the unknown observer error indicate the effect of the user-defined parameters of the observer, so that the guidelines for the choice of these parameters can be derived.

In summary, in this work an observer is proposed for online estimation of the unknown state, but also: (i) the bound of the steady state of the observer error for the unknown state is determined as a function of the model error and the user-defined observer parameters; (ii) the bound of the trajectory of the observer error is determined as a function of the model error and the user-defined observer parameters; and (iii) the guidelines for choosing the observer parameters are significantly simpler than other common observer designs. In turn, these results allow us to achieve an improved online estimation of the unknown state: the setting of the observer parameters by the user is simpler, it considers the transient and steady state bounds of the observer error, and the model errors appearing in the dynamics of the known and unknown states are considered. In monitoring and process control tasks, these features lead to more accurate knowledge on the error of the estimate of the unknown state, so that the control parameters can be defined to achieve improved robustness.

## 4. Application of the Observer to Bioreactors: A Simple Bioreactor Model and Numerical Simulation

The proposed observer given by Equations (5) to (7) can be applied to estimate the reaction rate terms of bioreactors, for instance, specific growth rate or substrate uptake rate. To this end, simple bioreactor models are described as follows, and numerical simulations are given afterwards.

### 4.1. Simple Bioreactor Model

A generic fermentation model for bioreactors can be described by mass balance models of substrate, biomass and product concentrations. The bioreactor consists of a stirred tank of liquid volume ($v$), biomass concentration ($x$), substrate concentration ($s$) and product concentration ($p$). Additionally, a substrate solution of concentration $s_i$ is added at a rate $F_i$, in the case of fed-batch or continuous operation.

For the sake of simplicity, the cases of batch, fed-batch and continuous flow operation modes can be encompassed by a general mass balance model for constant density fermentation [35]:

$$\text{Volume}: \frac{dv}{dt} = F_i - F_0$$

$$\text{Biomass}: \frac{dx}{dt} = \mu x - Dx \tag{11}$$

$$\text{Substrate}: \frac{ds}{dt} = -\rho x + D(s_i - s) \tag{12}$$

$$\text{Product}: \frac{dp}{dt} = (\alpha\mu x + \beta x) - Dp$$

where $D = F_i/v$ is the dilution rate, $F_i$ is the feeding flow rate, $F_o$ is the outlet flow rate, $v$ is the broth volume, $s_i$ is the fed substrate concentration, $\mu$ is the specific growth rate; $\rho$ is the uptake rate, for which the expression

$$\rho = y_s\mu + m_s$$

Is commonly used, where $y_s$ is a yield coefficient, and $m_s$ is the maintenance coefficient.

In addition, $F_i = F_o = 0$ for batch mode; $F_i = F_o \neq 0$ for continuous operation mode; $F_i \neq 0$; $F_o = 0$ for fed-batch mode. Thus, $dv/dt = 0$ can be used for batch and continuous operation modes.

In case of continuous operation mode with a known biomass concentration and an unknown concentration of substrate, the growth rate $\mu$ can be estimated with the proposed observer. To this end, the biomass model (11) can be cast in the form (1), (2) with

$$x_1 = x;\ x_2 = \mu;\ h_1 = -\alpha Dx;\ b = x;\ h_2 = 0;\ \delta_2 = d\mu/dt$$

In case of continuous operation mode with known substrate concentration and unknown biomass concentration, the substrate consumption rate $\rho x$ can be estimated, recasting the substrate model (12) in the form (1), (2) with

$$x_1 = s;\ x_2 = \rho x;\ h_1 = D(s_i - s);\ b = -1;\ h_2 = 0;\ \delta_2 = d(\rho x)/dt$$

In case of continuous operation mode with known substrate and biomass concentrations, the specific substrate uptake rate $\rho$ can be estimated, recasting the substrate model (12) in the form (1), (2) with

$$x_1 = s;\ x_2 = \rho;\ h_1 = D(s_i - s);\ b = -x;\ h_2 = 0;\ \delta_2 = d(\rho)/dt$$

### 4.2. First Simulation Example

The formulated observer is applied for estimating the substrate uptake rate $\rho$ for a continuous bioreactor with known concentrations of substrate and biomass. The inlet concentration $s_i$ is inaccurately known: $s_i = s_{im} + \delta_{si}$, where $s_{im}$ is the known value of $s_i$, and $\delta_{si}$ is the uncertainty. The growth rate expression and model parameters are:

$$
\begin{aligned}
\mu = \mu_{max}\left(1 - \tfrac{x}{x_{max}}\right)^f;\ & \mu_{max} = 0.01484\ h^{-1};\ x_{max} = 0.31999\tfrac{g}{L};\ f = 1.607; \\
& \alpha = 1;\ y_s = 0.0234;\ m_s = 0.22425\ h^{-1}; \\
& s_{im} = 53\tfrac{g}{L};\ \delta_{si} = 0.1\,s_{im} \times \sin\left(\tfrac{2\pi}{\tau_{si}}t\right)\tfrac{g}{L};\ \tau_{si} = 1h \\
& x_{to} = 0.09\tfrac{g}{L};\ s_{to} = 49.6\tfrac{g}{L};\ D = 0.002\ h^{-1}
\end{aligned}
\tag{13}
$$

where the initial concentrations of biomass and substrate $x_{to}$, $s_{to}$ are positive; $\mu_{max}, x_{max}, f$ are coefficients of the specific growth rate. The model parameters $\mu$, $\mu_{max}$, $x_{max}$, $\alpha$, $y_s, m_s$, were obtained by numerical model fitting (not shown), for experimental data of a batch process of *Gluconacetobacter diazotrophicus* provided in [36]. The details of the experimental system and measurements are shown in [36] in pages 118–127. The substrate concentration is the known state, and the substrate uptake rate $\rho$ is the unknown state, so that model (12) can be cast in the form (1), (2) with

$$x_1 = s;\ x_2 = \rho;\ b = -x;\ h_1 = D(s_{im} - s);\ h_2 = 0;\ \delta_1 = D\delta_{si};\ \delta_2 = \frac{d\rho}{dt} \tag{14}$$

Additionally, the observer (5)–(7) provides the estimate of $\rho$; that is, $\hat{x}_2 = \hat{\rho}$. The observer structure is given in Figure 2a.

The following values of the observer parameters are used:

$$\varepsilon = 0.015;\ k = 10;\ \gamma = 72;\ \omega = 9;\ \hat{x}_{1|to} = x_{1|to} = 49.6\frac{g}{L};\ \hat{x}_{2|to} = 0\ h^{-1};\ \hat{\theta}_{to} = 0 \tag{15}$$

where $\varepsilon$, $k$, $\gamma$, $\omega$ are chosen in accordance with guideliness provided in Remark 5 and some trial-and-error effort.

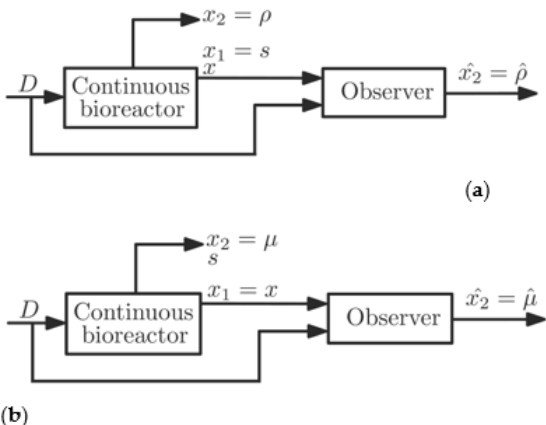

(a)

(b)

**Figure 2.** Structures of the observer applications to bioreactor: (**a**) first simulation example; (**b**) second simulation example. $x$ is the biomass concentration, $s$ is the substrate concentration, $\mu$ is the specific growth rate, $\rho$ is the specific substrate uptake rate, $D = F_i/v$ is the dilution rate, $F_i$ is the feeding flow rate, $v$ is the broth volume, $x_1$ is the known state, $x_2$ is the unknown state, $b$, $h_1$, $h_2$ are known terms of plant model (1), (2), and $\hat{x}_2$ is the estimate of $x_2$.

The observer simulation requires the observer (5)–(7), the plant model terms and parameters given by Equation (13), the definition of the terms of the system model given by Equation (14), and the values of observer parameters given by Equation (15). The simulations are shown in Figure 3.

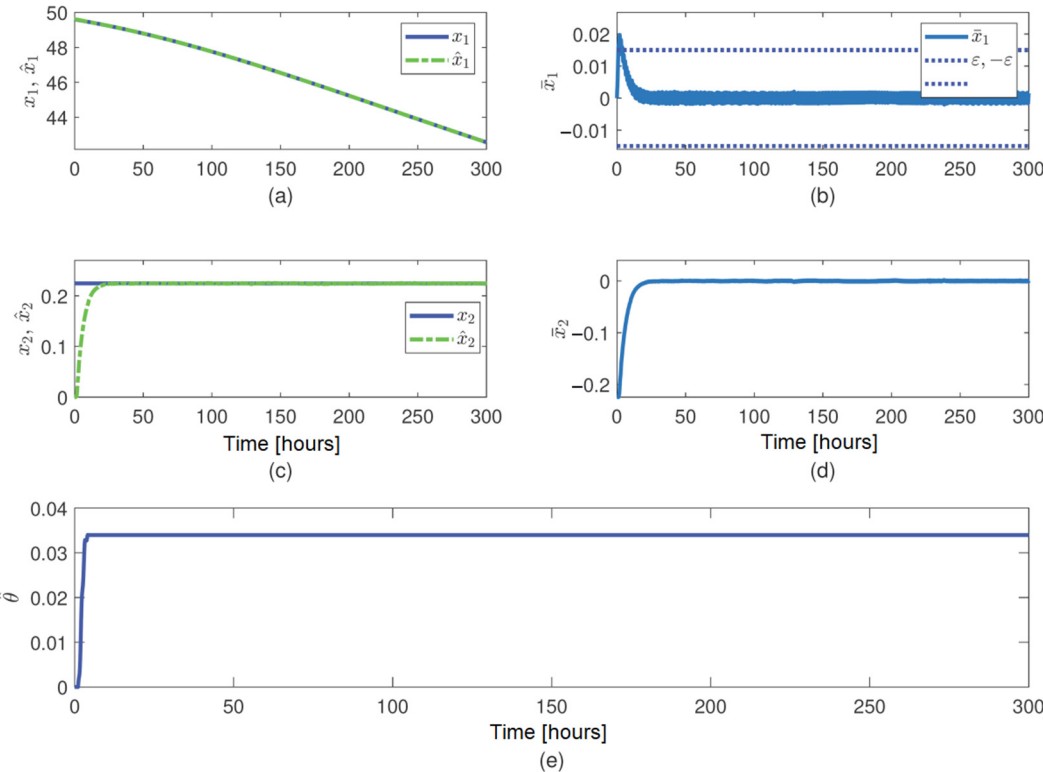

**Figure 3.** Performance of the proposed observer in the first simulation example: (**a**) trajectory of state $x_1$ and estimate $\hat{x}_1$; (**b**) trajectory of the observer error for the known state, $\bar{x}_1$; (**c**) trajectory of state $x_2$ and estimate $\hat{x}_2$; (**d**) trajectory of the observer error for the unknown state, $\bar{x}_2$; (**e**) trajectory of the updated parameter $\hat{\theta}_b$.

The observer simulations show that (Figure 3):

(i)    The observer error $\bar{x}_1 = \hat{x}_1 - x_1$ converges asymptotically to the compact set $\Omega_{x1} = [-\varepsilon, \varepsilon]$ and remains inside for $t \geq 3.7$ *h* approx (Figure 3a,b). Additionally, $\bar{x}_1$ exhibits an upward overshoot (Figure 3b), but the distance between the peak and the upper bound of $\Omega_{x1}$ is small.

(ii)    The observer error $\bar{x}_2 = \hat{x}_2 - x_2$ converges to its compact set in 18 h approx., and a low width of the convergence region is achieved, due to the small value of $\delta$ (Figure 3c,d). In addition, no overshoot is observed.

(iii)   The updated parameter $\hat{\theta}_\delta$ changes when $\bar{x}_1$ is outside the convergence set $\Omega_{x1}$, or equivalently when $\psi_{x1}$ is different from zero. This behavior agrees with the update law (7), which is a function of $\psi_{x1}$ (Figure 3e).

In addition, the choice of the user-defined parameters of the observer is quite simple, involving only some trial-and-error. The used values allow to cope with both uncertainties $\delta_1$ and $\delta_2$.

### 4.3. Second Simulation Example

The developed observer is employed to estimate the specific growth rate $\mu$ for a continuous anaerobic digester with a known concentration of biomass, whose biomass model is [37]:

$$\frac{dx}{dt} = \mu x - \alpha D x \tag{16}$$

$$\frac{ds}{dt} = -k_1 \mu x + D(s_i - s) \tag{17}$$

here, $\alpha$ is the biomass fraction in the liquid phase; $x$ is the concentration of acidogenic bacteria; $s$ is the concentration of chemical oxygen demand (COD). The fraction $\alpha$ is inaccurately known: $\alpha = \alpha_m + \delta_\alpha$, where $\alpha_m$ is the known value of $\alpha$, and $\delta_\alpha$ is the uncertainty. The growth rate expression and model parameters are:

$$\mu = \mu_{max}\frac{s}{s+K_s}; \ \mu_{max} = 1.2 \ d^{-1}; \ K_s = 7.1\frac{g}{L};$$
$$\alpha_m = 0.5; \ k_1 = 42.14; \ s_i = 10\frac{g}{L};$$
$$x_{to} = 0.3\frac{g}{L}; \ s_{to} = 1.2\frac{g}{L}; \ D = 0.35 \ d^{-1} \tag{18}$$
$$\delta_\alpha = 0.37 \ \alpha_m \times \sin\left(\frac{2\pi}{\tau_\alpha}t\right); \ \tau_\alpha = 3.5 \ d$$

where the initial concentrations of biomass and substrate, $x_{to}$, $s_{to}$, are positive and $\mu_{max}$, $K_s$ are coefficients of the specific growth rate. The details of the experimental system, measurements, and model, including parameters and specific growth rate expression, are given in [37]. The biomass concentration ($x$) is the known state, and the specific growth rate $\mu$ is the unknown state, so that model (16) can be cast in the form (1), (2) with

$$x_1 = x; \ x_2 = \mu; \ b = x; \ h_1 = -\alpha_m D x; \ h_2 = 0; \ \delta_1 = -\delta_\alpha D x; \ \delta_2 = \frac{d\mu}{dt} \tag{19}$$

Additionally, the observer (5)–(7) provides the estimate of $\mu$, that is, $\hat{x}_2 = \hat{\mu}$. The observer structure is given in Figure 2b. The following values of the observer parameters are used:

$$\varepsilon = 0.015; \ k = 10; \ \gamma = 72; \ \omega = 9; \ \hat{x}_{1|to} = x_{1|to} = 0.3\frac{g}{L}; \ \hat{x}_{2|to} = 0 \ d^{-1}; \ \hat{\theta}_{to} = 0 \tag{20}$$

The observer simulation requires the observer (5)–(7), the plant model terms and parameters given by Equation (18), the definition of the terms of the system model given by Equation (19), and the values of observer parameters given by Equation (20).

The simulation of the observer shows that (Figure 4):

(i)    The observer error $\bar{x}_1 = \hat{x}_1 - x_1$ converges asymptotically to the compact set $\Omega_{x1} = [-\varepsilon, \varepsilon]$ and remains inside for $t \geq 1.25$ *d* approx. (Figure 4a,b). Additionally,

$\overline{x}_1$ exhibits a downward overshoot, but the distance between the peak and the lower bound of $\Omega_{x1}$ is small.

(ii) The observer error $\overline{x}_2 = \hat{x}_2 - x_2$ converges to its compact set in 5.6 days approx., and a low width of the convergence region is achieved due to the small value of $\delta$ (Figure 4c,d). In addition, there is a downward overshoot (Figure 4d), but it is small when measured in respect to $\overline{x}_{2|to}$.

(iii) The updated parameter $\hat{\theta}_\delta$ changes when $\overline{x}_1$ is outside the convergence set $\Omega_{x1}$, or equivalently, when $\psi_{x1}$ is different from zero. This behavior agrees with the update law (7), which is a function of $\psi_{x1}$ (Figure 4e).

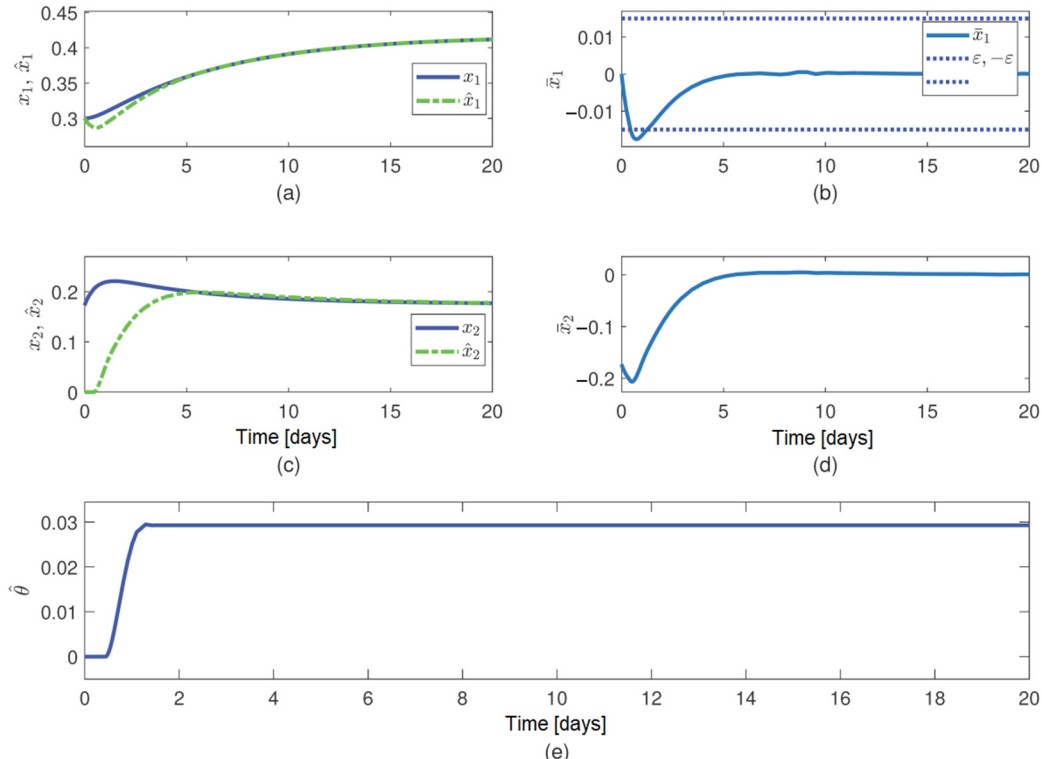

**Figure 4.** Performance of the proposed observer in the second simulation example: (**a**) trajectory of state $x_1$ and estimate $\hat{x}_1$; (**b**) trajectory of the observer error for the known state, $\overline{x}_1$; (**c**) trajectory of state $x_2$ and estimate $\hat{x}_2$; (**d**) trajectory of the observer error for the unknown state, $\overline{x}_2$; (**e**) trajectory of the updated parameter $\hat{\theta}_b$.

In addition, the choice of the user-defined parameters of the observer is quite simple, involving only some trial-and-error. The used values allow us to cope with both uncertainties $\delta_1$ and $\delta_2$.

### 4.4. Third Simulation Example

The developed observer is employed to estimate the specific growth rate $\mu$ for a continuous microalgae bioreactor, based on known concentrations of biomass and substrate, using the Droop model ([37]):

$$\frac{dx}{dt} = \mu x - Rx - Dx \tag{21}$$

$$\frac{ds}{dt} = -\rho x + D(s_i - s)$$

$$\frac{dq}{dt} = \rho - \mu q$$

where $x$ is the biomass concentration, $s$ is the substrate concentration, $q$ is the cell quota of assimilated nutrient; and $\rho$ is the specific substrate uptake rate. The growth rate expression and model parameters are:

$$\mu(q) = \max\left\{0,\ \mu_m\left(1 - \frac{Q_O}{q}\right)\right\}; \ \rho = \rho_m\left(\frac{s}{s + K_s}\right);$$

$$\rho_m = 0.03\frac{mgN}{mgC \cdot d}; \ K_s = 0.0010\frac{mgN}{L}; \ \mu_m = 0.5\ d^{-1}; \quad (22)$$

$$Q_O = 0.045\ mgN/mgC;$$

$$x_{to} = 0.1\ mgC/L; \ s_{to} = 0.01\ mgN/L; \ q_{to} = 0.06\ mgN/mgC;$$

$$D = \begin{cases} 0.25\left(1 + \sin\left(\frac{2\pi}{\tau_D}t\right)\right)d^{-1} & \text{for } t < 6\ d \\ 0 & \text{for } t \geq 6\ d \end{cases};$$

$$\tau_D = 8\ d; \ s_i = 0.05\ mgN/L;$$

In addition, $R$ is inaccurately known: $R = R_m + \delta_R$, where $R_m$ is the known value of $R$, and $\delta_R$ is the uncertainty; $R_m = 0.081$; $\delta_R = 0.1R_m \times \sin\left(\frac{2\pi t}{\tau_R}\right)$; $\tau_R = 3$. The details of the model, including parameters and specific growth rate expression are given in [2]. The biomass concentration ($x$) is the known state, and the specific growth rate $\mu$ is the unknown state, so that biomass model (21) can be cast in the form (1), (2) with

$$x_1 = x; \ x_2 = \mu; \ b = x; \ h_1 = -R_m x; \ h_2 = 0; \ \delta_1 = -\delta_R x; \ \delta_2 = \frac{d\mu}{dt} \quad (23)$$

Additionally, the observer (5)–(7) provides the estimate of $\mu$, that is, $\hat{x}_2 = \hat{\mu}$. The following values of the observer parameters are used:

$$\varepsilon = 0.007; \ k = 40; \ \gamma = 200; \ \omega = 9; \ \hat{x}_{1|to} = x_{1|to} = 0.1\ mgC/L; \ \hat{x}_{2|to} = 0\ d^{-1}; \ \hat{\theta}_{to} = 0 \quad (24)$$

The observer simulation requires the observer (5)–(7), the plant model terms and parameters given by Equation (22), the definition of the terms of the system model given by Equation (23), and the values of observer parameters given by Equation (24).

The performed simulations confirm the adequacy of the parameter recommendations provided in Remark 5 to achieve proper convergence speed and width of the convergence region of $\bar{x}_2$: a low value of $\varepsilon$; a value of $\hat{x}_{1|to}$ in the range $\left[-\varepsilon + x_{1|to},\ \varepsilon + x_{1|to}\right]$; a high value of $\gamma$; and a high value of $k$. The observer error $\bar{x}_1$ converges faster than $\bar{x}_2$.

The simulation of the observer shows that (Figure 5):

- The observer error $\bar{x}_1 = \hat{x}_1 - x_1$ converges asymptotically to the compact set $\Omega_{x1} = [-\varepsilon, \varepsilon]$ and remains inside for $t \geq 8.95\ d$ approx. (Figure 5a,b). Additionally, $\bar{x}_1$ exhibits an upward and a downward overshoot, but the distance between the peak and the bounds of $\Omega_{x1}$ are small.
- The observer error $\bar{x}_2 = \hat{x}_2 - x_2$ converges to its compact set in 11.5 days approx., and a low width of the convergence region is achieved, due to the small value of $\delta$ (Figure 5c,d). In addition, there is a downward overshoot (Figure 5d), the width of which is significant compared to $x_2$ values, but it vanishes in 7 d approx.
- The updated parameter $\hat{\theta}_\delta$ changes when $\bar{x}_1$ is outside the convergence set $\Omega_{x1}$ (Figure 5e).

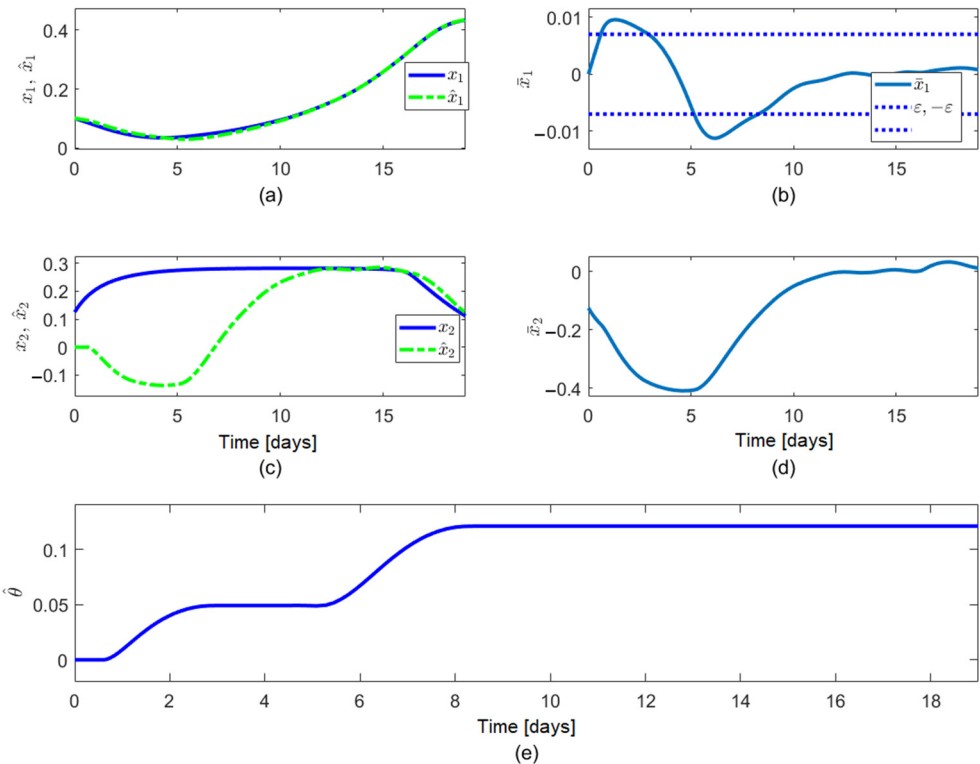

**Figure 5.** Performance of the proposed observer in the third simulation example: (**a**) trajectory of state $x_1$ and estimate $\hat{x}_1$; (**b**) trajectory of the observer error for the known state, $\bar{x}_1$; (**c**) trajectory of state $x_2$ and estimate $\hat{x}_2$; (**d**) trajectory of the observer error for the unknown state, $\bar{x}_2$; (**e**) trajectory of the updated parameter $\hat{\theta}_b$.

## 5. Conclusions

In this paper, a new observer design is proposed for second order systems applicable to generic fermentation models, considering bounded disturbance terms, and the dynamics of the unknown state are not required to be known by the observer. The bound of the transient response and the convergence region of the unknown observer error are determined in terms of the bounds of the disturbance, considering disturbances in the dynamics of both the known and unknown observer errors with a persistent but bounded nature. This is a significant contribution to closely related observer design studies, in which the transient response bounds are determined, but persistent and bounded disturbances are not considered in the dynamics of the known observer error. In addition, the guidelines for the choice of the observer parameters are provided. Other important contributions over current observer studies for systems with disturbances are: (i) the procedure for defining the observer parameters is greatly simpler, so that the solution to the Ricatti equation, solution to LMI constraints, and the accomplishment of eigenvalue inequality conditions are not required; (ii) discontinuous signals are not used in the observer; and (iii) the effect of the signum of the gain associated with the unknown state in the dynamics of the known state is explicitly and clearly considered in the design.

It was concluded that: (i) the upper bound of the transient response and the convergence region of the observation error of the unknown state depends on model terms and user-defined parameters; and (ii) the width of its convergence region can be reduced to some extent by properly defining the user-defined parameters, but it cannot be made arbitrarily small, due to the presence of the disturbance terms.

Numerical simulation shows that observer errors converge within a short time with a low estimation error, if observer-parameters are properly defined.

**Supplementary Materials:** The following supporting information can be downloaded at: https://www.mdpi.com/article/10.3390/fermentation8040173/s1. The detailed proofs of Theorem 1, Theorem 2 and Theorem 3 of the stability analysis are presented in the Supplementary material.

**Author Contributions:** Conceptualization, A.R.; methodology, A.R.; writing—original draft preparation, A.R., G.M.R. and F.E.H.; writing—review and editing, A.R., G.M.R. and F.E.H.; visualization, A.R., G.M.R. and F.E.H. All authors have read and agreed to the published version of the manuscript.

**Funding:** A.R. and G.M.R. were supported by Universidad Católica de Manizales. The work of F.E. Hoyos was supported by Universidad Nacional de Colombia—Sede Medellín.

**Institutional Review Board Statement:** Not applicable.

**Informed Consent Statement:** Not applicable.

**Data Availability Statement:** Not applicable.

**Acknowledgments:** This work was supported by Universidad Católica de Manizales and Universidad Nacional de Colombia, Sede Medellín. Fredy E. Hoyos thanks the Departamento de Energía Eléctrica y Automática. The work of Alejandro Rincón and Gloria M. Restrepo was supported by Universidad Católica de Manizales.

**Conflicts of Interest:** The authors declare no conflict of interest.

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
