# Peer review of "Design and Evaluation of a Robust Observer Using Dead-Zone Lyapunov Functions—Application to Reaction Rate Estimation in Bioprocesses"

_fermentation, doi:10.3390/fermentation8040173_

Round 1
Reviewer 1 Report
The manuscript presents an interesting approach for a state observer for bioprocesses. The paper fits into the scope of the Journal and presents the mathematic derivation of the underlying assumptions and equations for the observer. The manuscript is really interesting for those who want to control and automate bioprocesses or to build soft sensors for bioprocess monitoring. The mathematical derivation could be written more focused on the result and with fewer repetitions but the structure of the text is well.
The paper should be accepted after mayor revision. The following comments might help to improve the paper:
1 Introduction
Please refer to further state observers commonly used for bioprocesses, like kalman filter or the latest research findings on soft sensors.
- Application
Thank you for the detailed presentation of the implementation with some examples. Unfortunately, the data basis chosen for this is poorly selected, as the examples do not represent continuous cultivations and the experimental data are not accessible (FAIR data).
4.2 and 4.3 The growth rate and model parameters are not traceable out of the literature.
As I understood, the observer predict the steady state condition of a continuous cultivation. What is the added value of this approach compared to the pure model approach (dC/dt = 0) to calculate the steady state? A validation experiment for the result of the observer will be helpful.
5 Conclusion
Why is it an advantage that no discontinuous signals are used? However, in the case of the observer described here, no current process status is taken into account at all, neither discontinuous signals nor continuous measurements.
The fact that the growth rate depends on the dilution rate, or can be adjusted with it, is known for continuous cultivations. This can be directly calculated out of a process model with numerical fitted parameters. What is the benefit of your research? How can this approach contribute to solving the challenges described at the beginning of the introduction (lack of knowledge, process optimization or process control)?
Author Response
Reply Reviewer # 1
Thanks to the reviewer #1 for his feedback.
I attach the response letter

Reviewer 2 Report
- The authors are suggested to rewrite the abstract section describing more precisely the key point.
- The language can be more simplified.
Author Response
Reply Reviewer # 2
Thanks to the reviewer # 2 for his feedback.
I attach the response letter.

Round 2
Reviewer 1 Report
The manuscript presents an interesting approach for a state observer for bioprocesses. The paper fits into the scope of the Journal and presents the mathematic derivation of the underlying assumptions and equations for the obsevrer. The manuscript is really interesting for those who want to control and automate bioprocesses or to build soft sensors for bioprocess monitoring.
Author Response
Thank you again for your manuscript submission:
The academic editor has some comments:
Before accepting the paper, please do the following corrections:
- Figures must be self-explanatory, so any abbreviations must be spelled-out
in the captions, Figures 1 and 2 do not spell-out any abbreviations/symbols
used
- Check axis labels in graphs, they must start with capital letter, write
"Time" instead of "time"
- Use formal writing, do not use first person (we, us, our, etc), please
correct
Reply: Thank you very much, I attach the paper with the corrections in green font
